# *APC*-Related Phenotypes and Intellectual Disability in 5q Interstitial Deletions: A New Case and Review of the Literature

**DOI:** 10.3390/genes14071505

**Published:** 2023-07-23

**Authors:** Flavia Privitera, Flavia Piccini, Maria Paola Recalcati, Silvia Presi, Silvia Mazzola, Paola Carrera

**Affiliations:** 1Laboratory of Clinical Molecular Genetics and Cytogenetics, IRCCS San Raffaele Scientific Institute, 20132 Milan, Italy; privitera.flavia@hsr.it (F.P.); piccini.flavia@hsr.it (F.P.); recalcati.mariapaola@hsr.it (M.P.R.); presi.silvia@hsr.it (S.P.); 2Medical Genetics, ASST del Garda, Desenzano, 25015 Brescia, Italy; silvia.mazzola@asst-garda.it; 3Unit of Genomics for Diagnosis of Human Diseases, IRCCS San Raffaele Scientific Institute, 20132 Milan, Italy

**Keywords:** 5q deletion syndrome, array-CGH, KCNN2, intellectual disability

## Abstract

The 5q deletion syndrome is a relatively rare condition caused by the monoallelic interstitial deletion of the long arm of chromosome 5. Patients described in literature usually present variable dysmorphic features, behavioral disturbance, and intellectual disability (ID); moreover, the involvement of the *APC* gene (5q22.2) in the deletion predisposes them to tumoral syndromes (Familial Adenomatous Polyposis and Gardner syndrome). Although the development of gastrointestinal tract malignancies has been extensively described, the genetic causes underlying neurologic manifestations have never been investigated. In this study, we described a new patient with a 19.85 Mb interstitial deletion identified by array-CGH and compared the deletions and the phenotypes reported in other patients already described in the literature and the Decipher database. Overlapping deletions allowed us to highlight a common region in 5q22.1q23.1, identifying *KCNN2* (5q22.3) as the most likely candidate gene contributing to the neurologic phenotype.

## 1. Introduction

Monoallelic interstitial deletions involving the middle portion of the long arm of chromosome 5 are relatively rare abnormalities. Spanning from cytobands 5q15 to 5q31, they have been extensively described in association with Gardner syndrome and Familial Adenomatous Polyposis 1 (FAP1; MIM#175100), autosomal dominant conditions due to inactivating variants of the APC regulator of the WNT signaling pathway gene *(APC,* MIM*611731) in 5q22.2 [1,2,3,4,5]. In addition to the increased risk of developing neoplasms, patients with *APC* interstitial deletions reported so far present specific clinical symptoms, and several attempts have been described to better define the most commonly associated phenotype [6]. Characteristic features comprise intellectual disability (ID), developmental delay, craniofacial dysmorphisms, and skeletal or renal anomalies; other less frequent features are bifid uvula [1], seizure disorder, deafness, refractory anemia [2], recurrent respiratory infections [3], and umbilical hernia [4]. The broad spectrum of clinical findings described to date suggests the involvement of several genes, which are still not better specified.

Although cancer predisposition has been remarkably described in its genetic causes, identifying the *APC* (MIM*611731) and the MCC Regulator of the WNT signaling pathway (*MCC*, MIM*159350) genes as the main ones responsible for the pathological phenotype [1,2,3,4,5], the genetic causes underlying ID in 5q interstitial deletions involving *APC* have never been investigated. In the present study, we report the case of a new patient suffering from intestinal polyposis, characterized also by ID and dysmorphic features. Analysis of a whole-genome array-comparative genomic hybridization assay (a-CGH) allowed us to highlight an interstitial 19.85 Mb deletion of chromosome 5q, encompassing several genes. Integration of the genomic (i.e., a-CGH and clinical exome sequencing) and clinical results of our patient with those so far reported in the literature and the Decipher database (DatabasE of Chromosomal Imbalances and Phenotypes using Ensembl Resources, https://www.deciphergenomics.org/, accessed on 1 December 2022) allowed for the identification of an overlapping region that may be associated with ID, suggesting, as a candidate for the neurologic phenotype, the potassium calcium-activated channel subfamily N member 2 (*KCNN2*, MIM*605879) gene, and furthering the current knowledge regarding 5q deletions phenotypes encompassing *APC*.

## 2. Materials and Methods

### 2.1. Human Subjects

The patient’s sample and signed informed consent arrived at the clinical laboratory of Molecular Genetics, IRCCS Ospedale San Raffaele (Milan, Italy), from the “ASST del Garda” hospital, Desenzano (Brescia, Italy). Genetic counseling was performed, and data on the patient’s family history and clinical and dysmorphic features were collected. After the detection of the 5q deletion, the genetic analyses were extended to the relatives closest to the patient, and all of them gave their written informed consent to the study, according to the declaration of Helsinki. 

Genomic DNA was isolated and extracted from EDTA peripheral blood samples using a Maxwell^®^ RSC 48 Instrument and the Maxwell^®^ RSC Whole Blood DNA kit (Promega, Madison, WI, USA), according to the manufacturer’s instructions. The DNA quantity was estimated using the Implen NanoPhotometer^®^ P360 v.2.0.0. (Implen, Munich, Germany). 

### 2.2. Genetic Testing

#### 2.2.1. Multiplex Ligation-Dependent Probe Amplification Assay (MLPA)

MLPA deletion/duplication analysis was performed on genomic DNA using the SALSA^®^ MLPA^®^ Probemix P043-E1 APC and reagent kit (MRC-Holland, Amsterdam, The Netherlands) according to the manufacturer’s protocol. Fragment separation was performed using an Applied Biosystems-3730 DNA Analyzer (Applied Biosystems, Waltham, MA, USA). Coffalyser.Net software (v.2205131739), provided by MRC-Holland, was used for data analysis. 

#### 2.2.2. NGS and Clinical Exome

NGS was performed using the TruSight Cancer Sequencing Panel (Illumina, San Diego, CA, USA) which covers 94 high-risk genes associated with cancer predisposition, including the *APC* gene. Later, library preparation for clinical exome was undertaken with the TruSight One Sequencing panel (Illumina, San Diego, CA, USA), targeting the exonic regions of >4800 genes. Both libraries were constructed and sequenced on the Illumina NextSeq550 (Illumina, San Diego, CA, USA) in accordance with the manufacturer’s instructions. Exome data were analyzed using the DRAGEN^TM^ Bio-IT (Dynamic Read Analysis for GENomics, Illumina), enGenome-eVai (CE-IVD, v.2.7), Alamut^TM^ Visual plus v.1.7 (Sophia Genetics, Lausanne, Switzerland, accessed on 1 December 2022), and IGV (Integrative Genomic Viewer, https://igv.org/, accessed on 1 December 2022) tools. The NCBI dbSNP (https://www.ncbi.nlm.nih.gov/snp/, accessed on 1 January 2023), 1000 Genome Project (https://www.genome.gov/27528684/1000-genomes-project, accessed on 1 January 2023), ClinVar database (https://www.ncbi.nlm.nih.gov/clinvar/, accessed on 1 January 2023), gnomAD (Genome Aggregation database, https://gnomad.broadinstitute.org/, accessed on 1 January 2023), LOVD v.3.0 (Leiden Open Variant Database, https://www.lovd.nl/, accessed on 1 January 2023), InSIGHT (https://www.insight-group.org/variants/databases/, accessed on 1 January 2023), and MasterMind^®^ (Sophia Genetics, https://www.genomenon.com/, accessed on 1 January 2023) databases were consulted for variants’ classification and interpretation.

#### 2.2.3. Whole-Genome Array-Based Comparative Genomic Assay (a-CGH)

A high-resolution whole-genome array-based comparative genomic hybridization (a-CGH) analysis was performed on the genomic DNA of the patient and their relatives, using the Cytosure Oligo array ISCA 4x180K OGT (Oxford Gene Technology, Oxford, UK), resolution ~50–60 Kb. Protocols provided by the suppliers were followed without modifications. The slide was scanned using the InnoScan 710 Microarray Scanner (Innopsys, Carbonne, France) and processed using Mapix^®^ software v.8.5.0 (Innopsys, Carbonne, France) with default settings. Analysis was performed using CytoSure OGT Interpret Software, v.4.11 (Oxford Gene Technology, Oxford, UK). The results included imbalances with at least four consecutive probes with abnormal log2ratios. The Database of Genomic Variant (DGV—http://dgv.tcag.ca/dgv/app/home, accessed on 1 December 2022), Decipher (DatabasE of Chromosomal Imbalances and Phenotypes using Ensembl Resources- https://www.deciphergenomics.org/, accessed on 1 December 2022), PubMed (https:// pubmed.ncbi.nlm.nih.gov, accessed on 1 December 2022), UCSC genome browser (https://genome.ucsc.edu, accessed on 1 December 2022), Database of Human CNVs (http://gvarianti.homelinux.net/gvariantib37/index.php, accessed on 1 December 2022), SFARI (Simon’s Foundation Autism Research Initiative) Gene Database (https://gene.sfari.org, accessed on 1 December 2022), and OMIM (Online Mendelian Inheritance in Man- https://www.omim.org/, accessed on 1 December 2022) databases were consulted and used in the interpretation of the results.

## 3. Results

### 3.1. Patient’s Clinical Description and Genetic Analyses

The proband is a 41-year-old man, the second male of non-consanguineous parents (Figure 1, family pedigree; subject II;4). He was first evaluated for intestinal polyposis and colon neoplasm at the “ASST del Garda” hospital, Desenzano (Brescia). The patient showed evident relational difficulties and considerable intellectual disability; he is in treatment at a rehabilitation institution for his daily life. Family history was positive for tumors on the paternal side: the father (Figure 1, subject I;1) died of a tumor (unspecified) and the paternal uncle had a gastric tumor too. Unfortunately, no other clinical data were available. The mother (Figure 1, subject I;2), on the other hand, had always had good health. None of the three siblings, apart from the sister (Figure 1, subject II;2) who suffered from epilepsy in childhood, reported any health, behavioral problems, or ID.

A closer physical examination of the patient showed developmental and speech delay, failure to thrive, considerable hypotonia, a high forehead, and malposition of the third molar (Table 1). The patient also showed head lipomas, a horseshoe kidney, and many kidney exophytic cortical cysts (Table 1). No other clinical findings were available, i.e., no IQ score, other neurologic examinations, or skeletal assessments.

Suspecting FAP1, analyses of NGS for a cancer-associated genes panel and specific MLPA for the *APC* gene were requested. The MLPA analysis showed an entire *APC* gene deletion (Figure 2A), which was then confirmed by a marked decrease in coverage in the gene obtained by NGS. Because of his severe ID and facial dysmorphisms, a contiguous gene deletion syndrome was suspected; clinical exome and a-CGH were therefore carried out. While the clinical exome showed no pathogenic variants, a-CGH highlighted a 19.85 Mb deletion in the region associated with 5q deletion syndrome (5q21.3q23.2, Figure 2B) and two other duplications in 5q23.3q31.1 (383 Kb) and 11p15.4 (187), evaluated as uncertain and likely benign, respectively. A segregation analysis performed on the mother and the four siblings revealed a de novo origin of the 5q21.3q23.2 deletion in the proband and established a maternal origin of the 11p15.4 duplication. The identification of the 5q23.3q31.1 duplication in two other siblings (Figure 1, subjects II;2 and II;3) suggests the paternal origin of the rearrangement, even if only inferred.

### 3.2. Molecular Characterization of the Candidate Region

The review of the clinical and genetic results shown by our patient with those so far reported in the literature and the Decipher database allowed us to focus on twelve 5q deletions (Table 1 and Table 2), spanning from 5q15 to 5q31.1 (Table 3). Copy number variant (CNV) breakpoints all refer to the Human Genome Assembly GRCh37/h19. Segregation analysis, which was not confirmed in most of the deletions reported in the Decipher database, showed a de novo origin of the deletion in our patient and all the ones described in the literature. Exceptions regard only Patient 394928 (Decipher database) and the one described by Hastings R.J. et al., 2000 (Case 1) [3] whose deletions had arisen from balanced parental rearrangements. Classical and molecular cytogenetic analysis showed additional CNVs for Patient 254731 (Decipher database), Patient 287397 (Decipher database), and the one reported by Lindgren V. et al., 1992 (Patient EC) [2], which have not been considered as involved in the onset of the phenotype and which have been re-evaluated to date as uncertain variants (Table 3). Deletions described by Lindgren V. et al., 1992 [2], Hastings R.J. et al., 2000 [3], Raedle J. et al., 2001 [4], and Ofner L. et al., 2006 [5], which were characterized by Southern, FISH, or microsatellite analysis, were all remapped on UCSC using the exact localization of the probe or the described short tandem repeats (STR) region in the genome to then define the breakpoints of the rearrangement (Table 3). The overlap of all the deleted regions allowed us to identify a minimal region for all the patients’ common clinical features of about 7.77 Mb, in cytobands 5q22.1q23.1, which contains 33 genes: *APC, DMXL1, YTHDC2, SEMA6A, KCNN2, CCDC112, DCP2, FEM1C, CAMK4, TMED7, NREP, TNFAIP8, PGGT1B, ARL14EPL, CDO1, SRP19, TSLP, TMED7-TICAM2, TICAM2, REEP5, WDR36, TSSK1B, TRIM36, SLC25A46, AP3S1, ATG12, STARD4, DTWD2, HSD17B4, COMMD10, MCC, LVRN,* and *EPB41L4A* (Figure 3; Table 4). Focusing on the neurological phenotype, seven genes were already reported as involved in the genetics of ID: *KCNN2*, *YTHDC2, SEMA6A, DCP2, CAMK4, AGT12,* and *EPB41L4A*. All of them are annotated in Table 4.

## 4. Discussion

Patients carrying 5q heterozygous deletions show variable phenotypes, not yet completely associated with specific genetic causes. The most common clinical features include predisposition to cancer, ID, dysmorphic facies, and neurodevelopmental delay. In the present study, we describe a new patient affected by severe ID, and showing a large interstitial 19.85 Mb deletion in 5q21.3q23.2 encompassing *APC*, and compare other similar deletions already reported in the literature in association with ID. The overlap of twelve deleted regions allowed us to define a common region associated with ID, enhancing the current knowledge by suggesting stronger candidate genes.

The new 5q22.1q23.1 overlapping region for ID includes 33 genes, all of which are listed in Table 4, showing firstly the ones more intolerant to loss-of-function variants and haploinsufficiency conditions. These prediction values are given by the pLI, LOEUF, and pHAPLO scores reported by the Decipher database. Among the deleted genes, seven were selected because of their involvement in ID and neurodevelopmental disorders, when mutated. In particular, *KCNN2* (*605879) encodes for a potassium calcium-activated channel subfamily N member 2, involved in membrane excitability. The gene is associated with myoclonic dystonia type 34 (#619724) and with neurodevelopmental disorder with or without variable movement or behavioral abnormalities (#619725), both with an autosomal dominant inheritance and possibly displaying different pathogenic mechanisms. For the myoclonic dystonia type 34, a missense variant was described by Balint B. and co-authors in a family reporting childhood-onset dystonia predominantly affecting the head and neck, with a fast tremor and superimposed myoclonus [11]. Intra-familial clinical variability was also described. This evidence was compatible with the phenotype of the dominant Kyoto spontaneous rats, but no conclusive results regarding the role of *KCNN2* in pathology have been assumed.

For the second trait associated with *KCNN2*, several pathogenic variants have been reported: nonsense variants, splice sites, exonic deletions introducing a premature termination codon, and missense variants affecting the conserved residues of the protein’s functional domains. All are described in patients with neurodevelopmental delay, mental impairment, and moderate-to-severe ID phenotypes [12]. All the described patients present de novo variants or those inherited from an affected parent, suggesting a pathogenetic mechanism due to the haploinsufficiency of the gene [12].

Concerning the other genes, in order of haploinsufficiency intolerance, *YTHDC2, SEMA6A, DCP2, CAMK4, AGT12,* and *EPB41L4A* might contribute to the observed ID phenotype. YTH domain-containing protein 2 (*YTHDC2*, MIM*616530) is a gene encoding for RNA helicases and it is involved in RNA processing and metabolism. It is reported in the SFARI database as a strong candidate gene for autism (score: 2). Moreover, rare de novo small variants and CNVs in the *YTHDC2* gene have been identified in individuals with ASD in the Asian population [7,8]. In particular, a duplication of approximately 796 Kb, inherited from a parent, is reported to partially disrupt the *YTHDC2* and *KCNN2* genes [8]. Considering the new pathogenetic scenario proposed by the current study, a probable involvement of both genes in the onset of the autistic phenotype cannot be excluded. Semaphorin 6A (*SEMA6A,* MIM*605885) encodes for a transmembrane semaphorin expressed in developing neural tissue, and knockout mice of Sema6a resulted in impaired neuronal connectivity [9,10]. Decapping mRNA2 (*DCP2*, MIM*609844) is a key component of an mRNA-decapping complex required for the removal of the 5-prime cap from mRNA [13]. The gene encodes for a protein involved in a delicate regulatory pathway, which can lead to autosomal recessive intellectual disability when altered [14].

The calcium/calmodulin-dependent protein kinase IV *(CAMK4*, MIM*114080) gene encodes an important mediator of calcium-mediated activity and dynamics, particularly in the brain, and it is presented as a strong candidate for autism in the SFARI database (score: 2). In the study described by Zech M. et al. [15], a de novo splicing variant introducing a premature termination codon was associated with intellectual disability and hyperkinetic movement disorder with a gain of function mechanism. Autophagy-related 12 (*ATG12*, MIM*609608) is a gene involved in cellular autophagy, and homozygous missense variants in this gene have been related to ataxia, developmental disorder, and mental retardation [16]. Finally, erythrocyte membrane protein band 4.1-like 4A (*EPB41L4A*, MIM*612141) encodes for a member of a protein superfamily thought to regulate the interaction between the cytoskeleton and plasma membrane, and it has been involved in autosomal recessive neurogenetic disorders through whole-exome sequencing (WES) studies [17].

Several genes could therefore contribute to the ID phenotype. Although, among them, *YTHDC2* and *CAMK4* are reported to be associated with pathological phenotype in an autosomal dominant pattern, only a few studies suggest this kind of transmission, not confirmed even in the OMIM database. Nevertheless, given their probable intolerance to haploinsufficiency, it is not possible to exclude their involvement in the ID phenotype. Based on current knowledge, the present study confirms *KCNN2* as the most likely autosomal dominant candidate gene for ID associated with the 5q deletions involving *APC*.

Concerning the main dysmorphisms and symptoms associated with this region, in all patients analyzed for the current study (sourced by the literature and the Decipher database), ID was reported in 100% (12/12); moderate-to-severe developmental delay and hypotonia in 50% (6/12); ear abnormalities in 58% (7/12); speech delay, gastrointestinal diseases, hypotonia, and failure to thrive in 42% (5/12); and micrognathia, skeletal abnormalities (including skull, bone, jaw, and finger anomalies) in 33% (4/12). Among other less common features, supernumerary teeth, a high-arched palate, and neck features were observed in 25% of the cases (3/12); a high forehead, hypertelorism, down-slanting palpebral fissures, and flat nasal bridge in 17% (2/12). Further apparently isolated clinical manifestations may include epilepsy, deafness, horseshoe kidney, umbilical hernias, renal or ovarian cysts, telangiectasias, bifid uvula, recurrent respiratory infections, and ophthalmological problems. Overall, our data confirm what is already reported in the literature in association with a large interstitial 5q deletion involving *APC*, strengthening the genotype–phenotype correlation with FAP1 but improving the association with ID.

It is well known that the genetics of ID is heterogeneous, with a thousand-plus mutated genes leading to ID, including very small aneuploid segments across all chromosomes. In general, many environmental and genetic factors can concur with the manifestation of this phenotypic trait, and the genetic causes may arise both from the perturbations of large gene networks or from the impairment of a main gene [18]. A certain variability can be observed just by considering chromosome 5q. Indeed, we took into account an interesting report about a patient with mild learning difficulties and a deletion beginning in 5q14.3, just proximal to the 5q15q31 region and partially overlapping (5q14.3q21.3) with it [19], and a second paper [20] considering the phenotypic variability of microdeletions in this region: some smaller ones having severe ID, some larger ones with milder phenotypes, making a genotype–phenotype correlation difficult. Based on these observations, the microdeletion phenotypes along the 5q15q31 region, displaying a marked clinical variability, are probably due to their encompassed genes and to the interactions with the patient’s genetic background relating to other chromosomes.

Nevertheless, the current study presents new genetic perspectives not previously considered for 5q deletions involving *APC*. This study focused on the genetic cause triggering ID in the 5q region, identifying *KCNN2* as the most plausible gene due to the association with a neurodevelopmental disorder accompanied by behavioral abnormalities, with a possible mechanism of haploinsufficiency. In our patient, excluding the already mentioned genes identified in the overlapping region, we analyzed all the genes predicted to be intolerant to haploinsufficiency, expressed in the brain and associated with ID; since exome analysis allowed us to exclude the presence of autosomal recessive pathogenic variants, we did not find any other candidate contributing to the ID phenotype.

An important limitation of the study was the poorness of clinical information regarding our patient. Actually, our patient was evaluated by a specialist for oncological genetic counseling; however, since the patient lives in a rehabilitation institution, it was not easy to meet and investigate him further, especially regarding the neurologic signs and, in particular, the dystonia-movement findings. From an oncological point of view, it is well known that deletions involving the *APC* gene are associated with classical FAP1, mainly characterized by polyps and colorectal cancer, and attenuated FAP1, with extracolonic manifestations, including polyps of the stomach, osteomas, dental abnormalities, benign cutaneous lesions, desmoid tumors, and others [21]. The present study confirms as recurrent features the presence of dental anomalies and highlights the genotype–phenotype correlation between *APC* gene variants and skin lesions, including lipomas (as described in our patient) [22]. Unfortunately, no clinical findings concerning changes in bone mass or an association with other types of tumors were available, not even in family history.

In conclusion, the study provides new clinical and molecular information useful for the management of cases with 5q deletions involving *APC*. Moreover, as previously mentioned, other deletions in 5q, as well as the ones in 5q14.3q21.3 encompassing the myocyte enhancer factor 2C (*MEF2C*, *MIM 600662) have already been reported in association with ID [19,20]. A better definition of the genotype–phenotype correlation can help clinical geneticists in managing patients affected by 5q deletions, leading to the involvement of different specialists to improve patient management and their clinical condition. Clinical exome analysis allowed us to exclude, in the non-deleted homologous chromosome 5, the presence of point mutations or indels in the genes associated with recessive phenotypes, identifying the deletion in 5q21.3q23.2 as the unique genetic cause of the patient’s phenotype. However, further studies are needed to better define the contribution and pathogenic mechanisms of the different genes involved.

## Figures and Tables

**Figure 1 genes-14-01505-f001:**
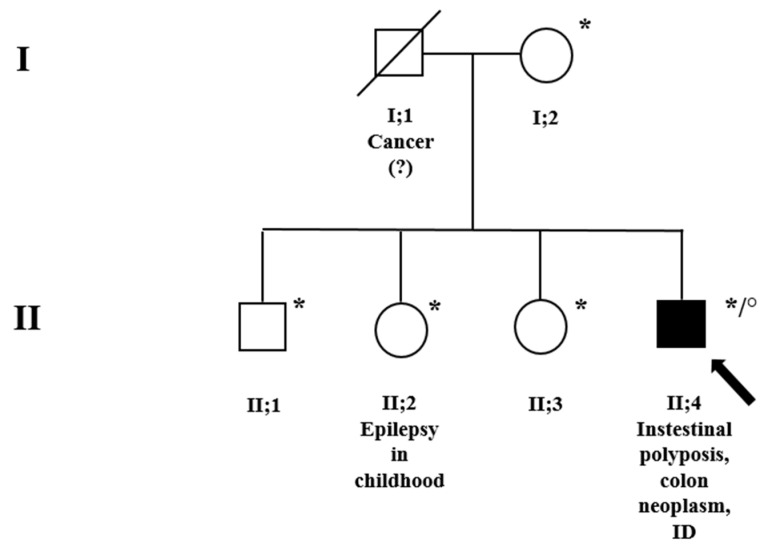
Family pedigree. * A-CGH was performed on these subjects. ° MLPA, NGS, and clinical exome were performed for the index subject, II;4, indicated by a black arrow. Squares = males; circles = females; and black = family members affected. Genetic counseling was conducted for the whole family.

**Figure 2 genes-14-01505-f002:**
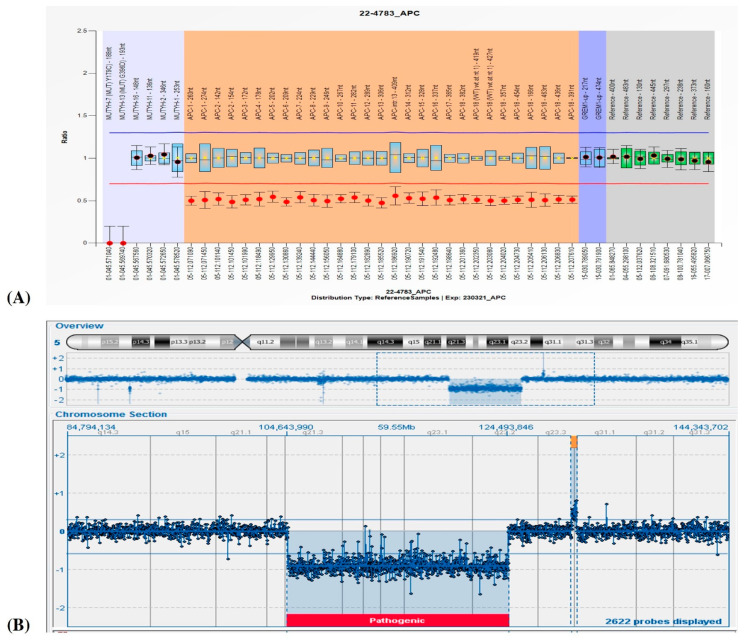
(**A**). Proband II;4: *APC* heterozygous deletion detected by MLPA (red dots at the 0.5 ratio value in the graph). (**B**). Heterozygous 5q21.3q23.2 deletion detected by a-CGH (CytoSure OGT Interpret Software, v.4.11).

**Figure 3 genes-14-01505-f003:**
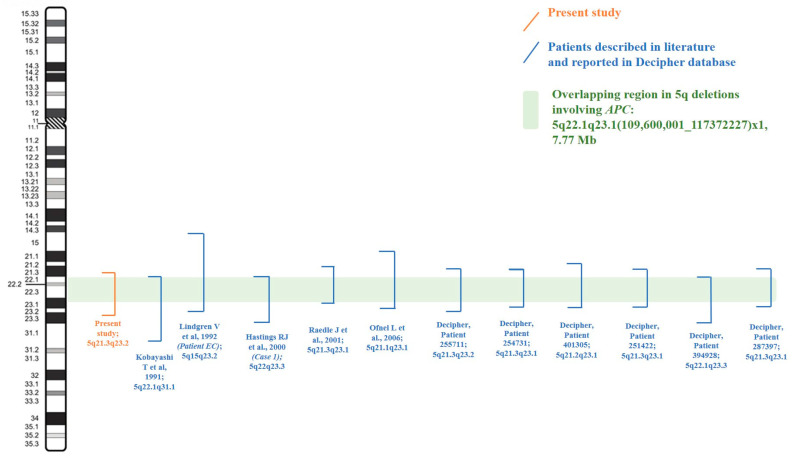
Chromosome 5 ideogram and overlapping region in 5q deletions.

**Table 1 genes-14-01505-t001:** Clinical findings associated with 5q deletions involving *APC*: proband from the present study and patients from the literature.

Clinical Features	Present Study	Kobayashi T. et al., 1991	Lindgren V. et al., 1992 (Patient *EC*)	Hastings R.J. et al., 2000 (*Case 1*)	Raedle J. et al., 2001	Ofner L. et al., 2006
Sex	M	M	F	M	M	F
Intellectual disability	+	+	+	+	+	+
Developmental delay (moderate/severe)	+	+	+, severe	+	-	-
Speech delay	+	+	+	n.a.	+	+
Failure to thrive	+	n.a.	+	+	+	+
Hypotonia	+	+	+	-	n.a.	+
High forehead	+	n.a.	n.a.	-	+	-
Ears abnormalities	-	+, low-set right ear	+, small and unusually folded	+, low-set	+	+, malrotated
Hypertelorism	-	+	n.a.	-	n.a.	+
Epicanthus	-	-	n.a.	-	-	+
Down-slanting palpebral fissures	-	-	n.a.	+	+	-
Flat nasal bridge	-	+	n.a.	-	-	+
Macrognathia	-	-	n.a.	-	-	-
Micrognathia	+	-	n.a.	+	+	+
Supernumerary teeth	+, malposition of a third molar	+	n.a.	-	-	+
Down-turned corners of the mouth	-	+	n.a.	-	n.a.	-
High-arched palate	-	+	n.a.	-	+	+
Neck features	-	n.a.	n.a.	+, short	+, short	+, wide
Skeletal abnormalities	-	-	+, metatarsus deformities	+, ulnar deviation of the fingers; polydactyly	+, long trunk, lumbar hyperlordosis	+, tapered fingers
Gastrointestinal diseases	+, colon neoplasm	+, Gardner syndrome; small polyp in the pyloric antrum	-	-	+, rectum colonic polyps; desmoid tumor	-
Familiarity with gastrointestinal diseases	+, gastric tumor in the paternal uncle	-	-	-	-	-
Others	Head lipomas; horseshoe kidney; kidney exophytic cortical cysts	Bifid uvula; atrophy of the right optic nerve	Seizure disorder not better specified; deafness; polycystic ovaries; refractory anemia	Recurrent respiratory infections	Umbilical hernia	Telangiectasias on the back and the ears

“+”: present; “-”: absent. F: Female; M: Male; and n.a.: not available.

**Table 2 genes-14-01505-t002:** Clinical findings associated with 5q deletions involving *APC* reported in Decipher Patients. Sanger–Decipher cases were omitted since their clinical descriptions are notoriously sparse.

Clinical Features	Patient 255711	Patient 254731	Patient 401305	Patient 251422	Patient 394928	Patient 287397
Sex	M	M	F	F	M	F
Intellectual disability	+	+	+	+	+	+, moderate
Short attention span	n.a.	+	-	n.a.	n.a.	n.a.
Autism	n.a.	+	-	n.a.	n.a.	n.a.
Growth abnormalities	n.a.	+	+, feeding difficulties	n.a.	n.a.	n.a.
Dysmorphisms	n.a.	n.a.	Low-set ears; brachycephaly; plagiocephaly; thin lower and upper lip vermilion; clinodactyly; hypotonia	n.a.	Chorioretinitis; long face; high anterior hairline; abnormal mandible morphology	n.a.
Spasticity	n.a.	n.a.	+	n.a.	n.a.	n.a.
Neoplasm of gastrointestinal tract	n.a.	n.a.	n.a.	+	+	n.a.

“+”: present; “-”: absent. F: Female; M: Male; and n.a.: not available.

**Table 3 genes-14-01505-t003:** Genetic findings obtained by classical and molecular cytogenetics. Breakpoints all refer to the Human Genome Assembly GRCh37/hg19.

Cases	Molecular and Cytogenetic Testing	Molecular Results (Start and Stop of the Deletions)	Location and Size of Deletions	Other Molecular Findings	Heredity of the Aberrations
Present study	A-CGH (4x180Kb)	104643990_124493846	5q21.3q23.2; 19.85 Mb	5q23.3q31.1(130278456_130661552)x3; 383 Kb, uncertain 11p15.4(5511754_5690132)x3 mat; 178 Kb, uncertain/ likely benign	5q21.3q23.2: de novo 5q23.3q31.1: unconfirmed parentage
Kobayashi T. et al., 1991 [1]	Karyotype	n.a.	5q22.1q31.1; ±38 Mb	-	De novo
Lindgren V. et al., 1992 *(Patient EC)* [2]	Karyotype; Southern hybridizations	92300001_121400001; involved probes: from C11p11 to MC5.61	5q15q23.2; ±35 Mb	dir ins(11)(p13q14.2q21)	De novo (both)
Hastings RJ et al., 2000 *(Case 1)* [3]	Karyotype	109600001_130600000	5q22q23.3; ± 21 Mb	-	Imbalance arising from a balanced parental rearrangement
Raedle J. et al., 2001 [4]	Karyotype; microarray (resolution not available) and microsatellite marker analysis	From STR D5S1466 to D5S639: 108919548_117372227	5q21.3q23.1; 8.45 Mb	-	De novo
Ofner L. et al., 2006 [5]	Karyotype; FISH (BAC clones) and microsatellite marker analysis	From clone RP11-203J7 to CTC-441N14: 102652547_121518358	5q21.1q23.1; 18.87 Mb	-	De novo
Decipher, Patient 255711	microarray (resolution n.a.)	108730123_124676977	5q21.3q23.2; 15.65 Mb	-	Unconfirmed parentage
Decipher, Patient 254731	microarray (resolution n.a.)	106886257_120800523	5q21.3q23.1; 13.91 Mb	7q21.3(95742135_95860080)x3; 117.9 Kb, uncertain	Unconfirmed parentage (both)
Decipher, Patient 401305	microarray (resolution n.a.)	103445913_119588365	5q21.2q23.1; 16.14 Mb	-	De novo (unconfirmed parentage)
Decipher, Patient 251422	microarray (resolution n.a.)	107429510_119168718	5q21.3q23.1; 11.74 Mb	-	De novo (unconfirmed parentage)
Decipher, Patient 394928	microarray (resolution n.a.)	109672101_127972101	5q22.1q23.3; 18.30 Mb	-	Imbalance arising from a balanced parental rearrangement
Decipher, Patient 287397	microarray (resolution n.a.)	101520750_118921525	5q21.3q23.1; 8.93 Mb	10p11.21(35100625_35530457)x3mat; 429.8 Kb, uncertain	5q21.3q23.1 del: de novo (unconfirmed parentage); 10p11.21 dup: maternal

De novo: present only in the proband; n.a.: not available; and -: absent.

**Table 4 genes-14-01505-t004:** Gene mapping in 5q21.2q22.3 region, listed according to their intolerance to haploinsufficiency.

Gene Full Name and Symbol (*OMIM)	Phenotype Number (#MIM)	Main Biological Activity	Inheritance	pLI ≥ 0	LOEUF~0	pHaplo ≥ 0.86	Association with ID (Yes/No)
*APC* (*611731) APC regulator of WNT signaling pathway	Familial Adenomatous Polyposis 1 (FAP1); Gardner Syndrome (#175100); Colorectal cancer, somatic (#114500); Gastric adenocarcinoma and proximal polyposis of the stomach (#619182); Gastric cancer (#613659)	Encodes a multidomain protein that plays a major role in tumor suppression by antagonizing the WNT.	AD	1	0.16	0.97	No
*DMXL1* (*605671) Dmx like 1	-	Encodes for a protein belonging to the superfamily of WD-repeat proteins, most of which have regulatory functions.	-	1	0.21	0.95	No
*YTHDC2* (*616530) YTH domain containing 2	-	Involved in RNA processing and metabolism.	-	1	0.13	0.88	Yes: reported in association with autism [7,8]. SFARI gene score: 2.
*SEMA6A* (*605885) Semaphorin 6A	-	Expressed in developing neural tissue.	-	1	0.22	0.98	Yes: described in axonal development and neural connectivity for brain functions, including cognition and memory [9,10].
*KCNN2* (*605879) Potassium calcium-activated channel subfamily N member 2	Dystonia 34, myoclonic (#619724); Neurodevelopmental disorder with or without variable movement or behavioral abnormalities (#619725)	Involved in calcium metabolism to potassium flux and membrane excitability.	AD	0.99	0.27	0.91	Yes: associated with dominant neurodevelopmental disorders [11,12].
*CCDC112* (-) Coiled-coil domain containing 112	-	-	-	0.85	0.87	0.38	No
*DCP2* (*609844) Decapping mRNA 2	-	Component of an mRNA-decapping complex required for the removal of the 5-prime cap from mRNA.	-	0.66	0.41	0.49	Yes: associated with intellectual disability; AR [13,14].
*FEM1C* (*608767) FEM1 Homolog C	-	The gene is moderately expressed in all adult and fetal tissues and specific brain regions; biological function is unknown.	-	0.63	0.48	0.54	No
*CAMK4* (*114080) Calcium/calmodulin-dependent protein kinase IV	-	Involved in neuronal transmission, synaptic plasticity, and the neuronal gene expression required for brain development and neuronal homeostasis.	-	0.51	0.47	0.36	Yes: once associated with neurodevelopmental disorder; AD [15]. SFARI gene score: 2.
*TMED7* (*619990) Transmembrane p24 Trafficking Protein 7	-	Belongs to the p24 family of small transmembrane proteins of the secretory pathway.	-	0.48	0.62	0.49	No
*NREP* (^*^607332) Neuronal regeneration-related protein	-	Induces myofibroblast transformation in a TGFB1-independent manner.	-	0.25	1.34	0.14	No
*TNFAIP8* (*612111) Tumor necrosis factor-α-induced protein 8	-	Its expression is induced by cytokines in human umbilical vein endothelial cells.	-	0.38	0.97	0.57	No
*PGGT1B* (*602031) Protein geranylgeranyltransferase type I subunit β	-	Involved in post-translational modifications of proteins, such as farnesylation.	-	0.04	0.60	0.42	No
*ARL14EPL* (-) ADP ribosylation factor-like GTPase 14 effector protein-like	-	Unknown	-	0.03	1.23	0.12	No
*CDO1* (*603943) Cysteine Dioxygenase	-	Metabolic function	-	0.01	0.93	0.43	No
*SRP19* (*182175) Signal recognition particle 19-KD	-	Assembles the SRP conserved ribonucleoprotein complex that mediates the cotranslational targeting of secretory and membrane proteins to cellular membranes.	-	0.01	1	0.27	No
*TSLP* (*607003) Thymic stromal lymphopoietin	-	Plays a key role in the development of B lymphocytes.	-	0	1.58	0.36	No
*TMED7-TICAM2* (-) TMED7-TICAM2 readthrough	-	-	-	0	0.8	0.33	No
*TICAM2* (*608321) TIR-domain containing adaptor molecule 2	-	Encodes for a protein involved in Toll receptor signaling.	-	0	1.6	0.2	No
*REEP5* (*125265) Receptor expression-enhancing protein 5	-	Involved in endoplasmic reticulum network formation.	-	0	1.27	0.21	No
*WDR36* (*609669) WD repeat-containing 36	Glaucoma 1 (#609887)	Involved in T-cell activation and is highly coregulated with IL12.	-	0	0.47	0.27	No
*TSSK1B* (*610709) Testis-specific serine kinase 1B	-	Belongs to a family of serine/threonine kinases highly expressed in the testis.	-	0	1.71	0.06	No
*TRIM36* (*609317) Tripartite motif-containing protein 36	Anencephaly 1 (#206500)	Expressed in the human fetal brain; biological function is unknown.	AR	0	0.67	0.29	No
*SLC25A46* (*610826) Solute carrier family 25, member 46	Neuropathy, hereditary motor and sensory, type VIB (#616505); Pontocerebellar hypoplasia, type 1E (#619303)	Belongs to the SLC25 family of mitochondrial carrier proteins.	AR	0	0.91	0.22	No
*AP3S1* (*601507) Adaptor-related protein complex 3, sigma-1 subunit	-	Part of a large complex (AP3) is involved in protein sorting.	-	0	1.6	0.27	No
*ATG12* (*609608) Autophagy-related 12	-	Involved in cellular cross-talks through the ATG12 conjugation system.	-	0	1.74	0.33	Yes: associated with ataxia, mental retardation, and developmental delay; AR [16].
*STARD4* (*607049) Start domain-containing protein 4	-	Involved in cholesterol homeostasis.	-	0	1.82	0.31	
*DTWD2* (-)	-	Unknown.	-	0	1.6	0.11	No
*HSD17B4* (*601860) 17-β-Hydroxysteroid dehydrogenase IV	D-bifunctional protein deficiency (#261515); Perrault syndrome type 1 (#233400)	Regulates the local cellular levels of estradiol.	AR	0	0.76	0.43	No
*COMMD10* (*616704) COMM domain-containing protein 10	-	Mediates protein–protein interactions.	-	0	1.87	0.13	No
*MCC* (*159350) MCC regulator of WNT signaling pathway	Colorectal cancer, somatic (#114500)	A tumor suppressor gene somatically rearranged in colon cancer tissue.	-	0	0.84	0.49	No
*LVRN* (*610046) Laeverin	-	Involved in implantation and placentation.	-	0	0.93	-	No
*EPB41L4A* (*612141) Erythrocyte membrane protein band 4.1-like 4A	-	Regulates the interaction between the cytoskeleton and plasma membrane.	-	0	1.01	0.25	Yes: associated with neurogenetic disorders and intellectual disability; AR [17].

AD: autosomal dominant; AR: autosomal recessive; and -: data not available. Whenever possible, SFARI gene scoring (https://gene.sfari.org/ accessed on 1 December 2022) is reported, classifying genes for their link to autism spectrum disorder (ASD) as S—syndromic; 1—high confidence; 2—strong candidate; and 3—suggestive evidence. pLI: probability of loss-of-function intolerance, based on the gene’s observed depletion of LOF variants in the Genome Aggregation Database (gnomAD, https://gnomad.broadinstitute.org/ accessed on 1 December 2022). Values range from 0 to 1. Genes with larger values (closer to one) are more intolerant of mutations. LOEUF: loss-of-function/expected upper bound fraction. A quantitative measure of the observed depletion (or enrichment) of loss-of-function variants in gnomAD compared to a null mutational model. The minimum value is 0, but there is theoretically no maximum value. Genes with smaller values (closer to zero) are more intolerant of mutations. pHaplo predicted probability of haploinsufficiency. This reflects the probability of haploinsufficiency (pHaplo; i.e., deletion intolerance) predicted from an analysis of large copy-number variants ascertained by microarrays in 950.278 individuals. pHaplo scores ≥ 0.86 indicate that the average effect sizes of deletions are as strong as the loss-of-function of genes known to be constrained against protein truncating variants (odd ratio ≥ 2.7); pHaplo scores ≥ 0.55 indicate an odds ratio ≥ 2. pLI, LOEUF, pHAPLO scores’ source: Decipher database (http://www.deciphergenomics.org/, accessed on 1 December 2022).

## Data Availability

Data sharing is not applicable to this article as no datasets were generated or analyzed during the current study.

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
