# Peer review of "APC-Related Phenotypes and Intellectual Disability in 5q Interstitial Deletions: A New Case and Review of the Literature"

_genes, 2023, doi:10.3390/genes14071505_

Round 1

Reviewer 1 Report

The authors, in this paper, presented a case report of a patient with a 19.85 Mb interstitial deletion identified by array-CGH, but also they compared this case with other patients carriers of deletions and phenotypes already described in literature and in the Decipher database. this study permitted to define a minimal critical region for mental retardation in 5q22.2q22.3, with KCNN2 (5q22.3) as the most likely candidate gene for autosomal dominant mental retardation.

The results are interesting because the study provides new clinical and molecular information useful for the management of cases with 5q deletions, as reported by the authors. The Methods are sound.

 I have minor issues to address:

 1. In the introduction section, page 2, the paragraph “Multiplex Ligation-dependent Probe Amplification assay (MLPA) showed initially a heterozygous deletion encompassing the entire ……Overlapping of the deleted regions allowed to identify a new minimal critical region for MR, suggesting new candidate genes never considered to date and furthering the current knowledge regarding 5q deletion syndrome phenotypes” seems to be a section of Materials and Methods. On the contrary, the authors did not specify the aim of their work, its added value and the originality.

2. The authors should insert a Limitations section.

3. to have a major uniformity I suggest (if pertinent) to link the Table 1 with Table 2 as the authors performed for the Table 3.

4. Check for the order of the Tables, for instance in the discussion section the authors should correct Table 3 (“The new 5q22.1q23.1 common minimal region for MR include 33 genes; all of them are listed in Table 3….”) with Table 4. Check also for this on the whole text.

5. Check for acronyms that should be reported in the extension form in the whole text. For instance, APC. Moreover, the authors should correct FAP1 instead of FAP.

Minor editing of English language required

Reviewer 2 Report

Dear authors

The paper "Minimal critical region for mental retardation involving KCNN2 in 5q Interstitial Deletion Syndrome: new case and review of the literature" by Flavia Privitera et al. was revised as requested previously.

We analyzed the originality, scientific quality, relevance to the field, presentation and adequacy of the references of the paper.

This manuscript is acceptable after major revision (discussion should be implemented and english language and style are fine/minor spell check required). Indeed, it is necessary to improve the resolution of the figures (since some of them are so unresolved that it is not possible to visualize the results) and include more recent references on the subject.

In addition:

-Figures 3A and 3B appear to be cropped and need to be redone.

-The authors should reorganize table 3 because it is not with a good presentation.

-The article has only 16 references and that number should be implemented, as well as should include more recent references on the subject.

English language and style are fine/minor spell check required.

Reviewer 3 Report

This article could be an excellent addition to the cytogenetic literature as its patient description and DNA analysis is excellent and its review of similar cases potentially useful if more focused. The idea that there is a critical region for mental retardation (intellectual disability please) is misguided as it is well-documented that a thousand-plus genes can have mutations leading to ID including very small aneuploid segments across all chromosomes. The authors could profit by reading our articles below, the first describing a patient with mild learning differences and a deletion beginning just proximal (5q14.3) to their 5q15q31 region of review but overlapping (5q14.3q21.3) considerably with it. The second shows the phenotypic variability of microdeletions in this region, some smaller ones having severe ID while larger ones like ours have milder phenotypes. The authors should recognize that there will be a series of microdeletion phenotypes along thes 5q15q31 region, each one variable due to its encompassed genes and interactions with patient background genes from other chromosomes. Common polygenic/multifactorial phenotypes like ID arise from perturbations of large gene networks and can occur with any significant chromosome change, hence no critical regions.

Instead, the authors should focus on their patient's 5q22.2/22.3  APC gene region, rarer cancers and anomalies able to have more narrow predisposing regions although with variable expression based on the extent of aneuploidy and influence of patient background. Gather only reports concerning this narrow region and focus on the question of how haploinsufficiency of the KCNN2 (associated with dystonia 34--OMIM 619724--and a neurodev disorder with or without movements OMIM619725) and APC genes (APC inhibiting Wnt, associated with multiple tumors, Wnt-1 associated with a form of osteoporosis OMIM 615221) might relate to the findings in their patient and others with deletions of the 5q22/23 region.

The WES result can be described as normal without specification of the WES method and the Sanger-Decipher cases omitted or listed only as 5q22/23 microdeletions since their clinical descriptions are notoriously sparse.  The patient description should be greatly expanded, documenting the types of tumors in his family members and including more detailed neurologic and dysmorphology examinations complete with IQ scores and activities of daily living competencies. Presence of neurologic and particularly dystonia-movement findings plus those of bone density in the patient should be emphasized.  

Tonk, V, Kyhm JH, Gibson CE, Wilson GN. Interstitial deletion 5q14.3q21.3 with MEF2C haploinsufficiency and mild phenotype: when more is less. Am J Med Genet 155A:1437-41, 2011.

Wilson GN. The MEF2C Gene-Microdeletion 5q14.3 Dilemma and Three Axioms for Molecular Syndromology. Am J Med Genet 158A:1589–1593, 2013.

English is excellent with some awkward phrases but always should be judged by how well most British and Americans would write in Italian

Round 2

Reviewer 1 Report

The authors answered my questions exhaustively. The paper is now eligible for publication.

Reviewer 2 Report

Dear authors

The paper "Minimal critical region for mental retardation involving KCNN2 in 5q Interstitial Deletion Syndrome: new case and review of the literature" by Flavia Privitera et al. was revised as requested previously.

We analyzed the originality, scientific quality, relevance to the field, presentation and adequacy of the references of the paper.

This manuscript is acceptable in present form.